

# Reconstructing the potential configuration in a high-mobility semiconductor heterostructure with scanning gate microscopy

Gaëtan J. Percebois[1], Antonio Lacerda-Santos[2], Boris Brun[2],
Benoit Hackens[3], Xavier Waintal[2] and Dietmar Weinmann[1*]

**1** Université de Strasbourg, CNRS, Institut de Physique et Chimie des Matériaux
de Strasbourg, UMR 7504, F-67000 Strasbourg, France
**2** PHELIQS, Université Grenoble Alpes, CEA, Grenoble INP, IRIG, Grenoble 38000, France
**3** IMCN Institute, NAPS Division, UCLouvain, Chemin du Cyclotron 2,
Louvain-la-Neuve B-1348, Belgium

* Dietmar.Weinmann@ipcms.unistra.fr

## Abstract

The weak disorder potential seen by the electrons of a two-dimensional electron gas in high-mobility semiconductor heterostructures leads to fluctuations in the physical properties and can be an issue for nanodevices. In this paper, we show that a scanning gate microscopy (SGM) image contains information about the disorder potential, and that a machine learning approach based on SGM data can be used to determine the disorder. We reconstruct the electric potential of a sample from its experimental SGM data and validate the result through an estimate of its accuracy.

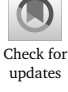

## 1  Introduction

During the past decades, nanostructured two-dimensional electron gases (2DEGs) in high-mobility semiconductor heterostructures have become a staple of condensed matter experiments [1, 2]. It is then crucial to have the best possible knowledge of their physical properties in order to perform predictive simulations of the electronic properties of such systems [3]. However, the precise disorder potential remains unknown and leads to unpredictable sample-to-sample fluctuations. In high-mobility modulation-doped heterostructures, it is expected that the main contribution to the disorder stems from the electrostatic Coulomb potential of ionized dopants [4]. The positions of those dopants vary from sample to sample, and the different disorder configurations are at the origin of fluctuations in the quantum transport properties of samples with the same macroscopic properties [5–7].

In order to determine the disorder potential seen by the electrons of a two-dimensional electron gas in such samples, we propose a method that combines scanning gate microscopy (SGM) and deep learning analysis. SGM is an experimental technique in which the conductance of a sample is measured while a biased Atomic Force Microscope tip scans over its surface [8].[1] The electrostatic potential of the tip locally modifies the potential landscape seen by the electrons and leads to a change in the sample conductance that depends on the position of the tip. The resulting spatially resolved conductance data contain rich information about the electronic transport in the two-dimensional electron gas. Experimental SGM data of systems containing a quantum point contact (QPC) show the emergence of a branching pattern in the conductance maps, ascribed to the branching of electrical current flow. The origin of this phenomenon was ascribed to microscopic focusing mechanisms related to the details of the disordered electrostatic background potential [9, 10]. While mean quantities like the correlation length of the SGM response depend on the tip strength [11], the branch structure and the whole SGM response depend crucially on details of the disorder potential [12]. Therefore, one can expect that the SGM response contains information that allows us to determine the disorder potential seen by the electrons.

Theoretical approaches to the quantum transport of nanostructured systems allow computing numerically the electronic quantum transport properties once the disorder potential is known. An example is the widely used software package KWANT [13] that is based on a tight-binding lattice approach. To determine the disorder potential from the transport properties is then an inverse problem that can be tackled using machine learning approaches. A related

---

[1]Scanning tunneling microscopy (STM), where a current from the sample to the tip allows extracting the local density of states, cannot be applied to usual heterostructures because of an insulating top layer.

idea has appeared recently consisting in a proof of principle where the disorder is determined in Majorana nanowire systems [14]. Another application which consists in using machine learning to adjust device parameters to compensate for uncontrolled disorder effects has been recently implemented in the case of a double quantum dot nanostructure [15]. It has also been suggested that properties of the disorder between the fingers of a QPC can be extracted from SGM data using cellular neural networks [16] or a swarming algorithm [17].

Our approach uses deep neural networks that are trained with simulated SGM-response data to yield the disorder potential in a two-dimensional electron gas in which electrons are injected through a QPC. We validate the found potentials and provide a quantitative estimate of their accuracy by comparing results for different QPC transmissions. Our work builds on previous researches [18] that have provided a proof of principle concerning the possibility to extract disorder information from quantum transport properties of a QPC system. In that work, the fast-to-compute but not directly measurable partial local density of states (PLDOS) has been used as input, a quantity that presents similarities with the SGM response [19]. The trained neural networks could reproduce the disorder potential of simulated test samples with high accuracy. In the present paper, we extend that approach to use SGM response data as input, we apply the trained neural networks to experimental data, and we estimate the accuracy of the resulting disorder potential.

The interest of our study is based on the absence of a direct experimental method to determine the disorder potential. However, this leads to important difficulties. One of them is that the disorder of an experimental device is unknown and it is not possible to use experimental data to train our neural network. We thus have to use simulated training data. To generate a training dataset, we generate many different random configurations of the disorder potential and compute the corresponding quantum transport properties. Since the quality of the prediction depends strongly on the quality of the training data, we made a special effort to determine the electric potential seen by the electrons by solving the quantum electrostatic problem self-consistently [20]. Furthermore, we fixed the macroscopic sample parameters to the values of the sample of Ref. [21], for which SGM-response data are available. Another difficulty arising from the impossibility to directly measure the disorder in a sample is the lack of a direct verification of the disorder prediction of our neural network when fed with experimental SGM data. In this paper, we propose a method to quantify the quality of the prediction, based on observations made with simulated data and on comparisons of disorder predictions for different QPC transmission values.

Our paper is organized as follows: The experimental parameters of the chosen sample and the model used to reproduce the real sample are described in Sec. 2. The simulation of the training data based on that model is detailed in Sec. 3. The application of the trained neural network to the experimental data and the evaluation of the reliability of the results is presented and discussed in Sec. 4. After the conclusions in Sec. 5, appendices offer detailed information on the implementation of the electrostatic environment (App. A), the learning procedure (App. B), the used neural network (App. C), and the estimation of the precision of the potential prediction (App. D).

## 2 System description

### 2.1 Experimental conditions

This paper is dedicated to the disorder potential prediction of a two-dimensional electron gas in a heterostructure based on SGM measurements. For illustration, we apply our study to a paradigmatic type of nanostructured 2DEG containing a QPC. Within the applications of the

SGM technique that are available in the literature, this type of nanostructure is the most widely studied one.

In order to apply our machine learning method, we have to create training data from simulated samples (i.e. simulated disorders and their associated SGM response). As it will be detailed later, the numerical generation of samples takes a considerable amount of time. Therefore, we choose for our study samples that can be simulated in a reasonable amount of time. The most obvious restriction concerns the size of the sample that has to be small enough. The temperature is another parameter that has an important impact on the computation time, since high temperature implies to compute the transmission factor at different energies.

The chosen experimental sample consists in a QPC defined on top of a $Al_{0.3}Ga_{0.7}As/GaAs$ heterostructure, that has been measured in Louvain-La-Neuve [21]. The 2DEG is located in the GaAs layer 67 nm below the following stacking of layers (from the surface of the sample to the 2DEG):

- Hafnium oxide 10 nm.
- GaAs 7 nm (capping).
- AlGaAs undoped 5 nm.
- AlGaAs 15 nm (Si doped $[4.8 \times 10^{24}\,\mathrm{m}^{-3}]$).
- AlGaAs 30 nm (spacer).

The 2DEG inside the heterostructure has an electron density of $n_{\mathrm{s}} = 2.53 \times 10^{15}\,\mathrm{m}^{-2}$ and a mobility of $\mu = 3.25 \times 10^{6}\,\mathrm{cm}^{2}\,\mathrm{V}^{-1}\,\mathrm{s}^{-1}$. The Fermi wavelength has been experimentally determined from the periodicity of the interference fringes in the SGM maps, yielding about $\lambda_{\mathrm{F}} = 40$ nm. A similar value $\lambda_{\mathrm{F}} = \sqrt{2\pi/n_{\mathrm{s}}} = 49.8$ nm is derived from $n_{\mathrm{s}}$ with the assumption of a parabolic dispersion. The QPC is realized by applying a negative voltage on two metallic gates that have a width of 150 nm, an opening of 250 nm and a thickness of 105 nm. SGM scans have been performed at four different values of the QPC gate voltage that are such that the unperturbed conductances (i.e. without the tip) of the sample are equal to $1.0\,G_0$, $1.3\,G_0$, $1.7\,G_0$ and $2.0\,G_0$, where $G_0 = 2e^2/h$ is the conductance quantum. The SGM scan has been performed in a cryostat with a base temperature set to 100 mK. The metallic tip used for the SGM is placed 30 nm above the heterostructure[2] and has a curvature radius of 50 nm. The voltage applied on the tip is maintained to $-6$ V during the scan, creating a depletion radius $r_{\mathrm{dep}} = 60$ nm. More details concerning the tip are presented in Ref. [22]. The scan is performed in a rectangular zone of dimension 520 nm × 260 nm located at 100 nm from the edge of the QPC. While the tip is scanning, the conductance is recorded by applying a 10 μV AC voltage excitation at 77 Hz. The current flowing through the device is recorded, simultaneously with the voltage drop in the device using two additional ohmic contacts.

## 2.2 Model of the sample

To model the experimental sample described in Sec. 2.1, we consider the heterostructure (depicted in the left panel of Fig. 1) as a stacking of four different layers. The first one is the substrate that is represented as a block of dielectric material with a relative permittivity $\epsilon_{\mathrm{r}} = 12$. Then, we have the 2DEG located at $z = 0$. The AlGaAs/GaAs stacking is represented as another block of dielectric material ($\epsilon_{\mathrm{r}} = 12$) where the dopants are represented by positive charges. Their positions are randomly chosen from a Poisson distribution[3] in a 15 nm thick region that

---

[2]Since the tip never touches the surface of the structure, piezoelectric effects are not expected to occur.

[3]We choose the independent random positions for the dopants while the dopant positions in a real sample are believed to be correlated [23]. However, for a sufficiently large number of samples, the whole configuration space should nevertheless be represented.

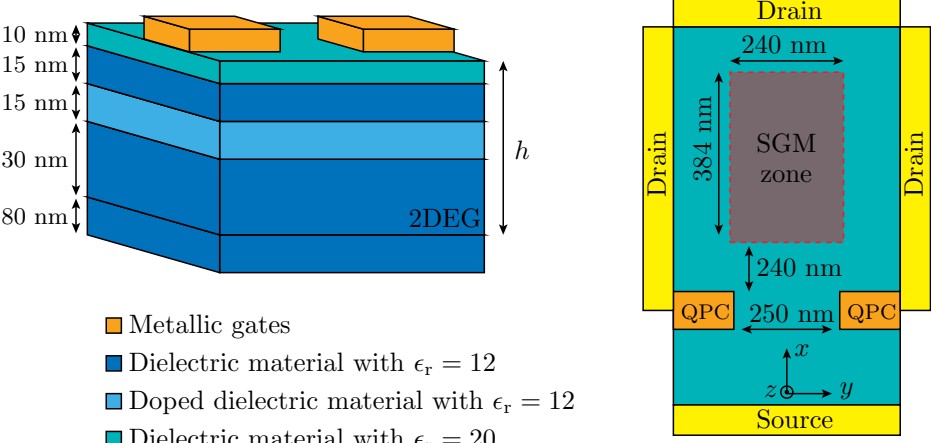

Figure 1: Sketch of the model that represents the experimental sample. The side view of the left panel highlights the layer stacking. The right panel depicts the top view of the sample and shows the location of the leads used for the quantum transport computations.

begins 30 nm above the 2DEG. Finally, the Hafnium oxide is represented by a layer with a dielectric permittivity of $\epsilon_r = 20$. All the electrostatic computations are performed using the Poisson Thomas-Fermi solver (PTFS) that is detailed in Sec. 3 and App. A.

The two metallic gates that create the QPC are represented by two cuboid elements with a width of 150 nm, a thickness[4] of 40 nm and a length such that they cover the entire lateral width of the sample except for an opening of 250 nm in the center. The tip is represented by a metallic half sphere with a radius of 50 nm and located 30 nm above the sample. To perform the scan, the tip moves in a plane parallel to the one of the 2DEG. The scanning zone for the simulations is a rectangle of 240 nm × 384 nm that begins 240 nm after the QPC. The latter zone has been reduced as compared to the available experimental data since the conductance is often very low when the tip is close to the QPC. This is due to the combined electrostatic effect of the tip and the QPC that closes the contact between the source and the drain.

In our simulation of the quantum transport, we use zero-temperature calculations as an approximation for the measurements at a very low experimental temperature. The leads are located as depicted in the right panel of Fig. 1. In order to avoid a discontinuity between the scattering region and the leads that can create reflection phenomena, we extend the edge of the sample by decreasing the potential smoothly to zero. It is also important to note that the quantum transport simulations are performed within an effective one-particle picture and do not take into account the electron-electron correlations beyond the mean field level. Such correlations are expected to become relevant at very low electron density, but they should not be important for the regime of experimental parameters considered in our study. Such an expectation is confirmed by the recent study of Ref. [3] where the behavior of a large number of samples was compared to simulated transport data.

## 3 Production of simulated training data

The generation of simulated sample data for the training of a neural network is performed in two steps. First we generate a random configuration of the dopants in the dopant layer,

---

[4]The thickness of the QPC gates does not impact the results.

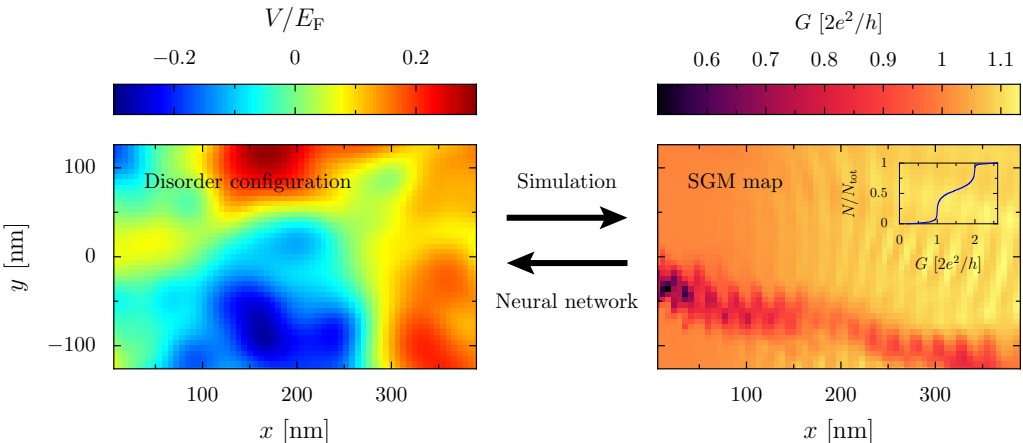

Figure 2: Example of training data. For one of the disorder configurations, the position-dependent potential is shown in the left panel, and the map of simulated SGM conductances is depicted in the right panel as a function of tip-position. A neural network is trained to solve the inverse problem, that is to take the SGM map as input and to predict the potential as its output. The inset of the right panel represents the cumulative distribution of unperturbed conductances for all samples of the dataset.

then we use the PTFS from the Python package called PESCADO [24] that solves the Poisson equation

$$\Delta U = \frac{\varrho}{\epsilon}, \tag{1}$$

and takes into account the density of states of the 2DEG within the Thomas-Fermi approximation. Once the self-consistent electrostatic potential is determined, we perform the electronic transport simulation with KWANT [13]. The discretization of the 2D space for the simulations is done with a lattice parameter $a_\parallel = 6\,\mathrm{nm}$ (see (A.2)) which leads to the ratio $\lambda_F/a_\parallel \approx 8$, large enough for reliable simulations. The SGM response consists of conductance values for various tip positions. In order to compute a full SGM conductance image, we therefore have to compute the electric potential created by the dopants, the QPC and the tip located at each of the considered positions. Thus, the number of potential maps that has to be computed for creating the entire dataset is equal to the product of the number of samples times the number of tip positions. Considering a dataset of a few thousand samples and even a low-resolution SGM response composed of a few thousand different tip positions, this sums up to require an enormous amount of numerical resources. The approximations detailed in App. A allow us to considerably reduce the number of electrostatic potential computations, while keeping a satisfying reliability. Moreover, considering the low temperature of the experimental sample, we perform the quantum transport computation at zero temperature, meaning that the conductance is computed only at $E = E_F = 9.1\,\mathrm{meV}$.

The correspondence between the simulation and the experimental sample is ensured by the following calibrations: the density of ionized dopants $n_{\mathrm{dop}}$, the voltage to apply on the tip $V_{\mathrm{tip}}$ and the QPC gate voltage $V_{\mathrm{QPC}}$. Considering that all the supplementary electrons of these dopants go to the 2DEG, the density of ionized dopants is $n_{\mathrm{dop}} = n_s/d_{\mathrm{dop}} = 1.69 \times 10^{23}\,\mathrm{m}^{-3}$, where $d_{\mathrm{dop}} = 15\,\mathrm{nm}$ is the thickness of the doping layer. The tip voltage is set to $V_{\mathrm{tip}} = -5.8\,\mathrm{V}$ such that the depletion radius is equal to the experimental one. The QPC gate voltage is chosen such that the conductance values $G$ without tip correspond to the experimental ones, which are located between the first ($G = G_0$) and the second ($G = 2G_0$) conductance plateau. The

conductance of a sample without tip varies with the disorder configuration. For the fixed QPC gate voltage that we choose, $V_{\text{gate}} = -0.82\,\text{V}$, most of the samples have a conductance around the first and the second plateau.

Using those parameters, we generated 2650 SGM-potential pairs. The resulting images have a resolution of $40 \times 64$ pixels. Using the symmetry of the problem with respect to the reflection at the $x$-axis (see Fig. 1), we double the number of training samples by reversing the SGM and the corresponding potential. Fig. 2 shows an example of such a potential (left) and the simulated SGM-response (right). The neural network is trained with those data pairs to solve the inverse problem, that is to predict the potential from the SGM-response. The statistics of the conductance of the generated samples in the absence of the tip is depicted in the inset of the right panel of Fig. 2 through the cumulative distribution function.

## 4 Prediction of the potential from an experimental SGM image

To obtain the potential prediction from an SGM image, we use a neural network trained with the data described in Sec. 3. Considering the complexity of the task, the number of SGM-potential pairs is relatively small. In order to improve the quality of the results in spite of the limited size of the available dataset, we use transfer learning. That method consists in a pre-training of the neural network to determine the potential from the partial local density of states (PLDOS) with a large number of training data. The pre-training dataset composed of PLDOS-potential pairs is significantly faster to create than SGM data, which allows us to generate a large dataset as described in App. B. After being pre-trained with PLDOS-disorder pairs, the network is trained with the SGM-disorder dataset to determine the disorder potential from the SGM response. Since both PLDOS and SGM share the same branching pattern for a similar disorder configuration (see Fig. 7), the two tasks of predicting the disorder potential from PLDOS and from SGM are similar, and the pre-training allows for significantly better results as compared to the usual random initialization of the neural network parameters. Of course, one can also improve the accuracy of the prediction by increasing the number of SGM-potential pairs that compose the training set. However, the dependence of the accuracy on the training set size presented in Fig. 9 shows that a considerable increase of the number of samples is needed to achieve a small improvement.

We quantify the performance of the completely trained network by the correlation between expected and predicted images $F^{(i)}$ and $F^{(j)}$ using the Pearson coefficient[5]

$$c_{ij} = \frac{\sum_p \left( F_p^{(i)} - \bar{F}^{(i)} \right) \left( F_p^{(j)} - \bar{F}^{(j)} \right)}{\sqrt{\sum_p \left( F_p^{(i)} - \bar{F}^{(i)} \right)^2} \sqrt{\sum_p \left( F_p^{(j)} - \bar{F}^{(j)} \right)^2}}, \qquad (2)$$

where the sums run over all $N$ pixels $p$ of the images, while $\bar{F}^{(i/j)} = \frac{1}{N} \sum_p F_p^{(i/j)}$ denotes the average value. On our test set, we obtain an average correlation between the expected and predicted potentials $\bar{c}_{\text{ep}} = 89\%$. While it can be expected that an increase of the size of the training set leads to a further improvement of that correlation (see App. C for quantitative details), we found no significant influence of the sample conductance on the performance of the neural network.

Using the previously described trained neural network combined with an additional data augmentation by smoothing the SGM response as discussed in Sec. 4.1, we are able to predict the potential associated to the four experimental SGM maps. The results are depicted in Fig. 3.

---

[5]This coefficient is equal to 1 in case of a perfect matching between the expected and predicted output and equal to zero when the two images are not correlated at all.

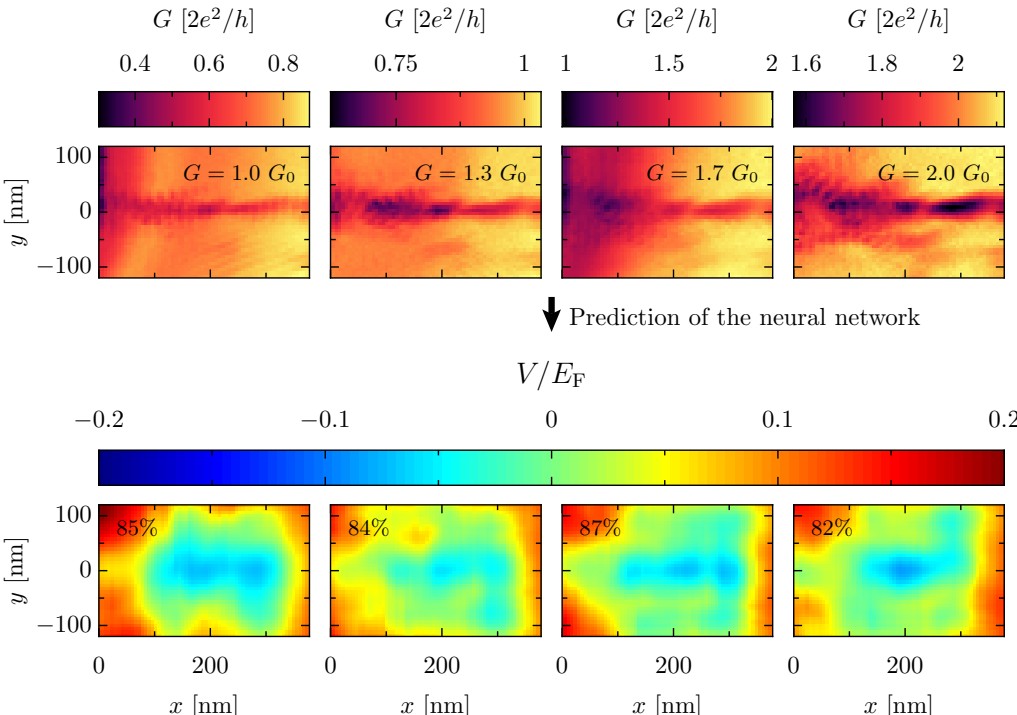

Figure 3: The top line represents the four experimental SGM maps obtained at the indicated QPC conductance values. The second line shows the corresponding predictions of the disorder potential obtained from the trained neural network when fed with the experimental SGM data as input. The estimated precision is indicated in the upper left corner of the predicted potentials (see Sec. 4.2). The SGM simulations corresponding to these predictions are shown in Fig. 12.

The estimated correlation between the predicted potential and the one actually present in the sample, indicated in the colorplots of the potentials, is determined through a method detailed in Sec. 4.2. Finally, an indirect evaluation method, applied to our sample in Sec. 4.3, confirms the validity of the predicted potentials.

## 4.1 Smoothing of simulated SGM response to approach experimental data

To perform the SGM experiment, a commercial AFM tip is glued to a tuning fork. During scanning, a small excitation is also applied to the tuning fork at its bare resonance frequency (32 kHz), resulting in a vibrating tip. Consequently, the depletion spot at the 2DEG level experiences a fast "breathing" motion, and the experimental SGM image in the end is a convolution of the different depletion spot radii. This could account for the blurring requiring the application of a Gaussian filter to reproduce the data.[6]

In order to take this issue into account, we train our neural network with a new data augmentation which works as follows: Before feeding the neural network with the simulated data to train it, we apply a Gaussian filter which corresponds to the convolution of the SGM image with a Gaussian bump of width $\sigma$ that is randomly chosen from a normal distribution with variance[7] $\eta_\sigma^2 = 23\,\mathrm{nm}^2$.

---

[6]The main source of difference between the simulation and the experiments in the SGM-response is the vibrating tip effect. To a lesser extent, the non-zero temperature, during the experimental measurements, is also responsible for the difference between the experiments and the simulations.

[7]The study that is used to determine the optimal variance is detailed in Ref. [25].

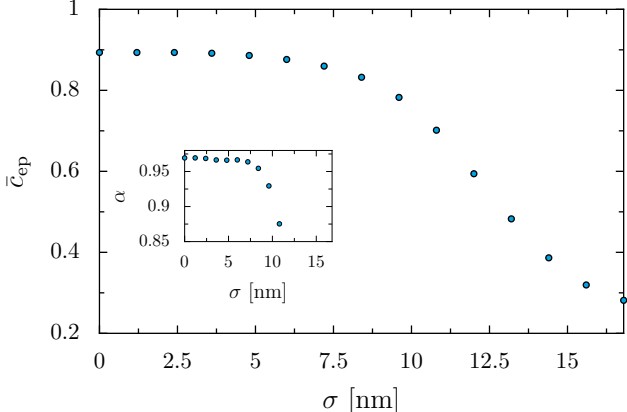

Figure 4: Average correlation between the predicted disorders with the exact one, obtained for the simulated test data with inputs blurred by a Gaussian filter with a width of $\sigma$. The inset represents the evolution of the coefficient $\alpha$ defined in Eq. D.1 as a function of the blur factor $\sigma$. The model used in this figure is the one that was used for the disorder predictions of Fig. 3.

While the application of a Gaussian filter on the simulated data allows us to get closer to the appearance of the experimental data, it seems important to evaluate the predictive power of the neural network on the simulated test set that has also been blurred with a Gaussian filter $\sigma$. Fig. 4 represents the evolution of the prediction performance with the parameter $\sigma$. The latter evaluations have been performed by averaging the Pearson correlation coefficient between the expected and predicted potential over all the samples of the test set. We notice that the neural network performs well on the test data while the blurring parameter remains below the value[8] of $\sigma = 7.5$ nm. Even if there is no obvious possibility to estimate the value of $\sigma$ that fits the best with the experimental data, a value of $\sigma$ that is equal or greater than 7.5 nm seems unlikely (see Fig. 11 in App. E).

## 4.2  Indirect evaluation of the prediction quality

When applying the neural network to an experimental SGM map to determine the disorder of a semi-conductor heterostructure, there is no possibility to compare the prediction to the real disorder. To get an estimate of the prediction quality in this most interesting case, we propose an indirect method to determine the reliability of the predicted disorder.

The indirect evaluation consists in the comparison of the predictions from several SGM scans that are obtained from the same sample under different experimental conditions. In our case, the varying parameter is the gate voltage of the QPC, with four different values that are available. The sample disorder, and thus the expected prediction from the SGM maps, should remain the same if the sample stayed in the cryostat at low temperature during the whole series of measurements. It is then possible to compute this Pearson correlation coefficient $c_{ij}$ between two predictions $i$ and $j$, as depicted in Fig. 5 for the example $i = 0$. We denote $\bar{c}_i = 1/3 \sum_{j \neq i} c_{ij}$, the average correlation of the sample $i$ with the others that can be used as an indicator for the quality of the predictions. When dealing with simulated data, we can also compute the correlation coefficient $c_{ie}$ between the predicted and expected potential. Therefore, based on simulated data, we establish a link between $c_{ie}$ and $\bar{c}_i$. The detailed study presented in App. D shows that in a given regime, one can obtain the linear relation $c_{ie} = \alpha \bar{c}_i$. A linear regression

---

[8]The drop observed in Fig 4 appears at lower values when the neural network is trained without the blurring data augmentation. This indicates that the blurring data augmentation has a significant impact on the final results [25].

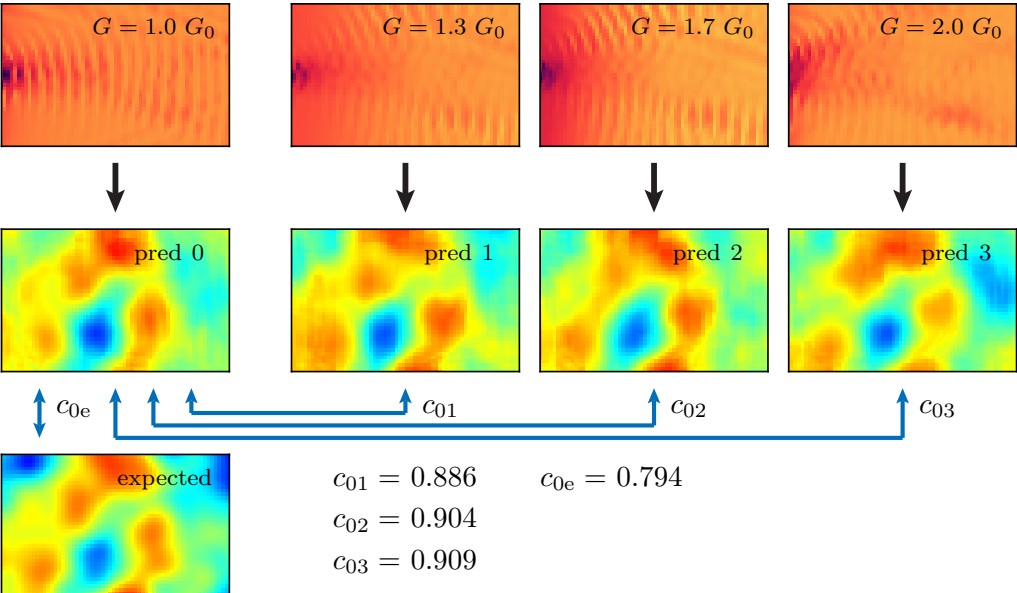

Figure 5: The four upper panels depict simulated SGM images of a sample with fixed disorder but under different gate voltages with the indicated unperturbed conductances. The black arrows represent the passage through the neural network. The four panels below the SGM images correspond to the potential predictions of the neural network. The lowest image is the exact disorder potential that corresponds to the four SGM images. For the example $i = 0$, the blue arrows link images among which are computed the Pearson correlation coefficients $c_{ij}$ and $c_{ie}$.

between the coefficients $c_{ie}$ and $\bar{c}_i$ computed from the prediction of simulated samples with different disorder configurations, as depicted in Fig. 10, yields $\alpha = 0.96$.

For consistency, we perform the same method to determine the proportionality coefficient $\alpha$ as the one used for Fig. 10, but for the test set inputs blurred as described in Sec. 4.1 with different values of the blurring parameter $\sigma$. The dependence of $\alpha$ on $\sigma$ is shown in the inset of Fig. 4. Considering that $\sigma \approx 6\,\mathrm{nm}$ (from Fig. 11 in App. E) is the value that seems to give the best similarity between experimental data and blurred simulated results, we find a value of $\alpha \approx 0.97$. Using this value of $\alpha$, we observe a distribution of the predicted Pearson coefficient around the real one, that can be fitted using Eq. D.2, with the parameters: $\zeta = 0.031$, $\omega = 0.050$ and $\theta = -1.167$ . This distribution is shown in [25].

## 4.3  Predicted potential from experimental SGM data

Having trained the neural network using the Gaussian filtering together with data augmentation and transfer learning methods described above, optimized and tested on the set of simulated data, we are finally in position to use it for the prediction of the disorder potential in the experimental sample of Ref. [21].

The experimental SGM-response (reshaped to the same resolution as the simulated ones[9]) is presented in the first line of Fig. 3. The disorder potential predictions made by the neural network are depicted in the second line of Fig. 3, with the estimates for the precision $\alpha\bar{c}_i$ in the upper left corners. The results with the highest correlation coefficient $\alpha\bar{c}_i = 0.87$ are the data associated to the conductance $G = 1.7\,G_0$.

---

[9]Due to the limitation of computational time, we have not been able to perform simulations with the same resolution as the one of the experiments.

Therewith, we finally estimate the correlation of the predicted potentials in the third column of Fig. 3 with the exact potential of the sample to about 87 %. We note that the variation between the predicted potential and the real one is of the order of the variation between the four predictions. Let us mention that the blurring data augmentation of the images described in Sec. 4.1 contributes considerably to the high fidelity of the prediction of the neural network. Without it, the best precision quantified by the coefficient $\alpha \bar{c}_i$ is around 67 % instead of the 87 % reached when using the blurring data augmentation.

Moreover, the reconstruction of the SGM-response from the predicted potentials and its comparison with the experimental data offers a qualitative way to validate the results of the neural network. Details are presented in App. F.

# 5 Conclusion and outlook

In this paper we have determined the disorder potential that is seen by the electrons inside a 2DEG in a semiconductor heterostructure. We have extended the machine-learning approach of Ref. [18] towards a prediction of the disorder from experimentally accessible SGM data, and we have used a self-consistent PTFS [20] in a particular effort towards a realistic simulation of real-life samples. To cope with the considerable increase in numerical effort related to realistic simulations of the SGM response, we have implemented data augmentation and transfer learning methods that allow us to train our neural networks with simulated SGM data for a reduced number of disorder realizations.

While our method provides a prediction of the unknown disorder potential, it is difficult to estimate the precision of that disorder without the knowledge of the exact potential landscape in a high-mobility modulation-doped heterostructure. We use a technique based on the comparison between the disorder predicted by the neural network from SGM data obtained for the same sample but at different experimental conditions to determine the quality of the prediction. Working with simulated data, where the exact disorder is known, we show that the similarity between the predicted potentials for different parameter values (evaluated through the Pearson correlation coefficient) is correlated with the similarity between the predicted and the expected disorder potential. This correlation allows us to get an estimate of the precision of the prediction based on the comparison of predictions for different experimental conditions that can be performed without knowing the exact disorder potential.

While the neural network trained with the raw simulated data already yields good results, a difference in sharpness between the experimental and simulated data, that might be due to the vibration of the tip during the measurement, led us to use a data augmentation method that consists in applying Gaussian filters to blur more or less strongly the simulated SGM images during the training process. From this improved training, we obtained a neural network able to predict the disorder potential of the experimental SGM with remarkable reliability. For four different openings of the QPC in the same sample, we obtained four similar disorder potential predictions. Using the above-mentioned accuracy estimation method (verified within the set of simulated data), we estimate the correlation between the predicted and the real disorder to be about 87 %. We also verified quantitatively the relevance of the prediction by computing the SGM from the predicted potentials, and obtained a good similarity between the branch pattern of the reconstructed SGM responses for the predicted disorder potentials and the experimental ones.

Using a limited amount of numerical resources, those results have been obtained within a certain number of approximations in our simulations while trying to be as close to reality as possible. The most important ones are probably the quantum transport computations performed for only one tip depletion radius, the reduced resolution of the simulated SGM re-

sponse and the superposition (A.1) that allowed us to not use the PTFS for each tip position in each disorder configuration. Without these approximations, the computational time would have been enormous. Nevertheless, with more numerical resources, a further increase of the precision of the determination of the disorder potential in semiconductor 2DEGs will be possible by improving the simulations on the mentioned aspects. Moreover, a better precision could be expected from data obtained in dedicated experiments, for example with a more homogeneous electron flow in the sample and with the variation of additional parameters like SGM tip strength and size or magnetic field.

The approximations made in the simulations induce the risk of a systematic bias contained in the training data that might then affect the predictions of the neural network that is trained to invert the KWANT simulation and to determine an electrostatic potential computed by PESCADO. Such a bias cannot be detected by testing the neural network on simulated data. However, the confidence in the reliability of the predicted disorder is increased by calculating sample properties based on the predicted potential, but for different experimental conditions, that can then be compared to measured data as in App. F.

The detailed knowledge of the disorder potential seen by the electrons in a nanostructured device, as it can be obtained with the method proposed in this work, can allow improving the comparison between experiment and theory beyond the qualitative level. Quantitative simulations including the precise disorder configuration can overcome the problematic sample-to-sample fluctuations and make it possible to predict the behavior of individual samples.

## Acknowledgments

We are indebted to Hermann Sellier, Ulf Gennser and Rodolfo Jalabert for useful discussions. We thank Philipp Weinmann for helpful tips on programming techniques.

**Funding information** This work of the Interdisciplinary Thematic Institute QMat, as part of the ITI 2021-2028 program of the University of Strasbourg, CNRS, and Inserm, was supported by IdEx Unistra (ANR 10 IDEX 0002), and by SFRI STRAT'US Projects No. ANR-20-SFRI-0012 and No. ANR-17-EURE-0024 under the framework of the French Investments for the Future Program. We gratefully acknowledge financial support from the French National Research Agency ANR through project ANR-20-CE30-0028 (TQT). B. H. (Senior Research Associate) ackowledges financial support by the F.R.S.-FNRS (PDR No. T.0105.21).

## A Approximation on the electrostatic environment

Due to the large amount of computer time required to create a full dataset including the self-consistent electrostatic potentials for all disorder configurations and tip positions, we use an approximation to significantly reduce the number of potential computations. The latter approximation consists in dividing the potential in two parts. We compute the potential that includes the disorder and the QPC gates $U_{\text{QPC,dis}}$. In parallel, we compute the potentials without disorder (i.e. for uniform dopant density) of the QPC and the tip for all tip positions $U_{\text{QPC,tip}}(\mathbf{r},\mathbf{r}_{\text{tip}})$. Still without disorder, we also compute the potential of the QPC alone $U_{\text{QPC}}$. Finally, the complete potential that includes the disorder, the QPC and the tip at a position $r_{\text{tip}}$ is then approximated as

$$U_{\text{full}}(\mathbf{r},\mathbf{r}_{\text{tip}}) = U_{\text{QPC,dis}}(\mathbf{r}) - U_{\text{QPC}}(\mathbf{r}) + U_{\text{QPC,tip}}(\mathbf{r},\mathbf{r}_{\text{tip}}). \tag{A.1}$$

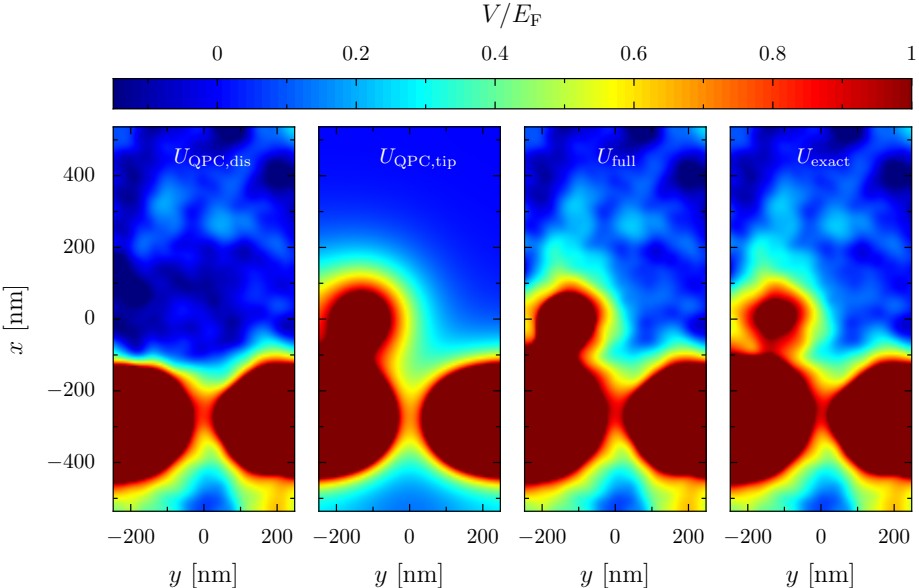

Figure 6: Illustration of the potential construction of Eq. (A.1). The two left panels are the potential parts obtained for the disordered sample with the QPC gate and for the clean sample with the QPC gate and the SGM tip, respectively. The resulting approximation $U_{full}(\mathbf{r}, \mathbf{r}_{tip})$ is shown in the third panel for a tip position $\mathbf{r}_{tip}$ close to the left QPC gate. The right panel depicts the exact potential computed for the same dopant distribution, in presence of all the elements (tip and QPC gates) at the same time.

In this way, we have to use the PTFS once for each sample, plus once for each tip position instead of using the PTFS for each tip position in every single sample. Since 69 PTFS in parallel take about 2600 seconds to compute, the approximation allows us to perform the dataset in 29 hours for all the tip positions and 28 hours for all the disorder configurations of the dataset. This results in a total processing time of 57 hours while the exact computation would have taken about 9 years.[10] The example given in Fig. 6 allows us to compare the results of the exact method and the approximation we make. We can observe minor differences. However, we note that for this example the tip is close to the QPC and thus corresponds to one of the worst cases for the approximation [25].

All the previously described potentials are computed using the PTFS PESCADO, using a discretization of space with lattice parameters parallel to the 2DEG $a_{\parallel}$, corresponding to the $x - y$ plane and perpendicular to the 2DEG $a_{\perp}$, which corresponds to the $z$ direction, with values

$$a_{\parallel} = 6\,\text{nm}, \qquad a_{\perp} = \begin{cases} 3z^2 + 2, & z < 0, \\ 5, & 0 < z < h + h_{QPC}, \\ 20, & z > h + h_{QPC}, \end{cases} \tag{A.2}$$

where $h + h_{QPC}$ corresponds to the thickness $h$ of the layers above the 2DEG plus the thickness of the gates $h_{QPC}$. Those values allow us to obtain results that are as accurate as possible while limiting numerical resources to a reasonable amount. We note that such a discretization does not allow us to reproduce the exact dimension of the layers described in Sec. 2.1 (The capping layer has been set to 5 nm instead of 7 nm which induces a small difference of 2 nm between

---

[10]The computations are performed using the cluster of X. Waintal's group, composed of 23 nodes, each with 48 cores and 128 GB of RAM. Only 6 to 7 cores have been used in each node because 7 jobs use all the 128 GB of RAM. Calculation times have been estimated using the PESCADO version dating from 18/07/2023.

the model and the real sample.). The parallel component of the lattice parameter has also been chosen by taking into account the convergence on the tight-binding conductance computation with KWANT.

The last tuning of the model to fit as well as possible to reality is the application of an offset voltage to the QPC gate to compensate the screening of the negative charges at the surface of the heterostructure. A uniform charge density is reached for $V_{\text{off}} = 0.162\,\text{V}$.

## B Transfer learning data

Since the amount of time required for the creation of the training data is a considerable issue in our study, we pre-train the neural network with data similar to the ones described in Sec. 3, but significantly faster to produce. In this appendix, we present the model used to produce the large pre-training dataset. The data production consists of two parts: the electrostatic model and the quantum transport model.

### B.1 PLDOS as input of the pre-training data set

The quantum transport computation is performed along the same lines as the one for the SGM response (lattice parameter, energy, position of the electrodes), but instead of computing the SGM response, we compute the partial local density of states (PLDOS) that arise from the source (i.e. the QPC) [26, 27]

$$\rho(\mathbf{r}) = 2\pi \sum_{a=1}^{N_{\text{qpc}}} |\psi_{\text{qpc},\epsilon,a}(\mathbf{r})|^2\,, \tag{B.1}$$

where $\psi_{\text{qpc},\epsilon,a}$ is the scattering wave function at energy $\epsilon = E_{\text{F}}$ that enters the 2DEG from channel $a$ of the QPC. Hence, this quantum transport information does not necessitate any tip to probe the 2DEG, and the full PLDOS image can be obtained in a single run of KWANT [13] instead of performing as many calculations as tip positions, which makes the PLDOS about 3000 times faster to compute than the SGM. A pre-training with PLDOS data makes sense because for the same disorder configuration, this signal has a similar branching pattern as the SGM response as depicted for an example in Fig. 7. Indeed, under some drastic condition the two signals can even be identical. However, those conditions (zero temperature, weak disorder, delta-shaped tip potential [19]) are not realistic and in practice one cannot apply directly a neural network trained with PLDOS on SGM images.

### B.2 Fast-to-compute potential

Again for computational issues, we approximate the full potential

$$U'_{\text{full}} = U'_{\text{dis}} + U'_{\text{QPC}}\,, \tag{B.2}$$

as the sum of the potentials of the disorder $U'_{\text{dis}}$ and the QPC gates $U'_{\text{QPC}}$. Since the potential-PLDOS relation is essential for the pre-training, and the relation to a realistic potential less relevant in this first step, we do not use the PTFS self-consistently for each tip position and disorder realization that takes a considerable amount of time (1 hour) compared to our super-position method (about 15 seconds).

The disorder potential is implemented following the approach of Ref. [4] where the sum of the electrostatic potentials of the ionized dopants is treated through a Fourier transform such that the sum is performed over a few hundred terms of the Fourier series instead of summing

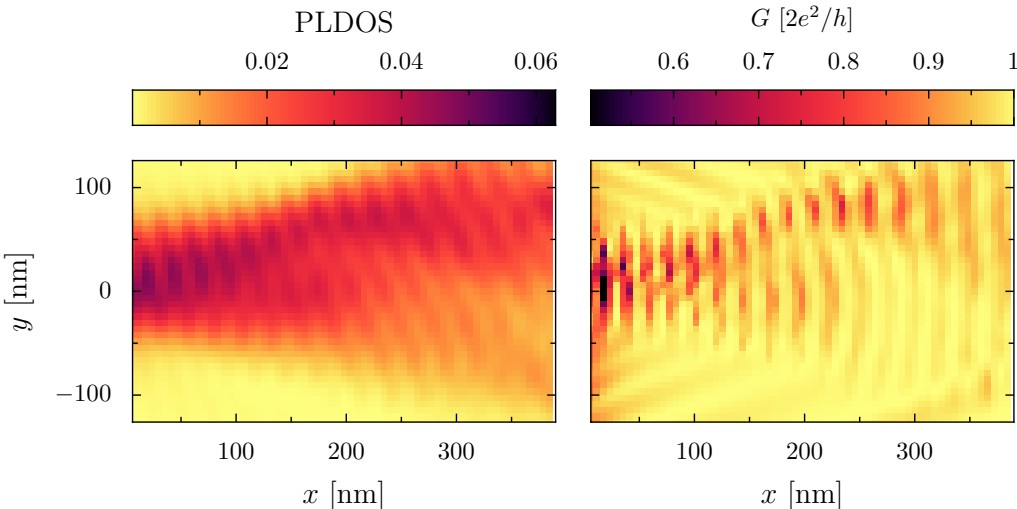

Figure 7: Example of PLDOS data (left) and the SGM response (right) computed for the same sample (i.e. the same disorder potential).

over the $10^4$ dopants. When considering a doping layer located at a distance $s$ of the 2DEG, the potential writes

$$U'_{\text{dis}}(\mathbf{r}) = -E^*_{\text{Ry}} a^*_{\text{B}} \frac{2\Delta q_x \Delta q_y}{\pi} \sum_{j \in \mathbb{R}^{+*}} \frac{e^{-q_j s}}{q + q_{\text{TF}}} r_j \cos\left(\mathbf{q}_j \mathbf{r} + \phi_j\right), \tag{B.3}$$

where $E^*_{\text{Ry}}$ and $a^*_{\text{B}}$ represent the effective Rydberg energy and Bohr radius, respectively. $\Delta q_y$ and $\Delta q_x$ correspond to the step-width of the reciprocal space labeled by the vector $\mathbf{q}$ of magnitude $q$. This model also takes into account the Thomas-Fermi screening through the term $q_{\text{TF}}$. The random position of the dopants is taken into account through the terms $r_j = \sqrt{\alpha_j^2 + \beta_j^2}$ and $\phi_j = \arctan\left(\alpha_j + i\beta_j\right)$, where $\alpha_j$ and $\beta_j$ are random numbers that follow a normal distribution centered in 0 and of variance $\sigma^2 = M/2$. $M$ is the number of dopants in the doping layer.

In order to represent the electrostatic potential created by the QPC gates (orange parts in Fig. 1), we use the model of Ref. [28], that defines the potential of a finite gate rectangle defined by $x_1 < x < x_2$ and $y_1 < y < y_2$ as

$$\frac{U_{\text{rect}}(x_1, x_2, y_1, y_2)}{A} = g\left(x - x_1, y - y_1\right) + g\left(x - x_1, y_2 - y\right)$$
$$+ g\left(x_2 - x, y - y_1\right) + g\left(x_2 - x, y_2 - y\right), \tag{B.4}$$

where $A$ is the potential value under the gate far from the edges and the function $g(u, v)$ is defined as

$$g(u, v) = \frac{1}{2\pi} \arctan\left(\frac{uv}{sR}\right), \quad \text{with} \quad R = \sqrt{u^2 + v^2 + s^2}. \tag{B.5}$$

The QPC gate potential is composed of two rectangular gates with edges at $x_1^{\text{left}}, x_2^{\text{left}}, y_1^{\text{left}}, y_2^{\text{left}}$ and $x_1^{\text{right}}, x_2^{\text{right}}, y_1^{\text{right}}, y_2^{\text{right}}$, such that

$$U'_{\text{QPC}} = U_{\text{rect}}(x_1^{\text{left}}, x_2^{\text{left}}, y_1^{\text{left}}, y_2^{\text{left}}) + U_{\text{rect}}(x_1^{\text{right}}, x_2^{\text{right}}, y_1^{\text{right}}, y_2^{\text{right}}). \tag{B.6}$$

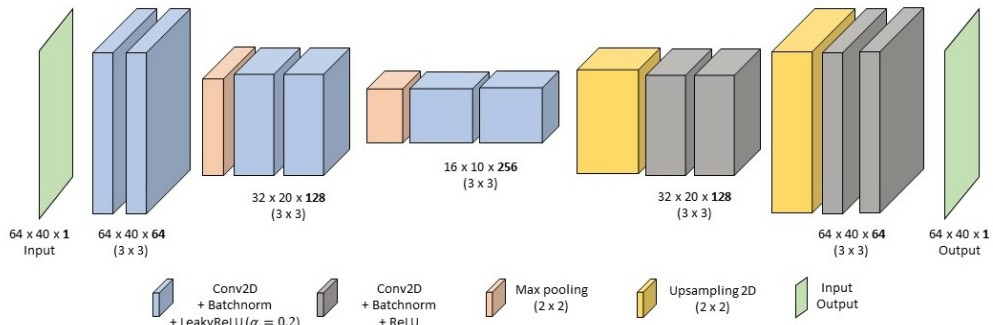

Figure 8: Sketch of the neural network architecture. All the layers are connected only to the previous layer. The dimension of the layers is indicated below them, where the two first numbers are the dimension of the image and the last one in bold corresponds to the number of feature maps.

## C   Neural network details

The neural network is first pre-trained with the data of App. B, which is composed of PLDOS-potential pairs. Computing the PLDOS (whole image) takes about the same time as computing the conductance for one position of the tip. Therefore, we created a first dataset of 50.000 samples to train a neural network to determine the potential from the PLDOS. The neural network being already trained to perform a task close to the wanted one, the main change in the final learning step is a shift of the input of the neural network from PLDOS to SGM.

Since the data are 2D arrays (for the input and the output), we use a neural network with a convolutional encoder-decoder structure. Such an architecture has the particularity to compress the information, keeping only the main features of the input image which are then used to perform the prediction. The convolutional architecture is broadly and successfully used for image analysis. The selected architecture is based on the ones commonly used to perform image translation. In order to fine-tune our model (i.e. architecture plus other parameters), we evaluated several dozens of models with the $k$-fold cross-validation method [29] over our 2450 samples. In our case, we chose $k = 8$ which is a compromise between good statistics and using a reasonable amount of computational resources. For models close to the one depicted in Fig. 8, we found that the batch size (the number of examples on which the neural network performs one training iteration) is one of the most important parameters. Testing many different values, we obtained the best results for rather small batch sizes.

The retained model from the cross-validation is the one depicted in Fig. 8. For the training on the SGM dataset, the size of the batch is 4, and we used the "binary cross-entropy" as loss function. Concerning the transfer learning, the only trainable layers were the two first convolutional layers. This model is tested on 200 samples that have not been used during cross validation. In order to quantify the accuracy of our model, we compute the Pearson correlation coefficient $c$. The distribution of correlations between the expected potential and the real one is depicted in the right panel of Fig. 9, where we can observe an asymmetric distribution with a mean correlation $\bar{c} = 0.90$. We can also observe from the inset of Fig. 9 that the accuracy of our model is independent of the conductance of the sample.

The left panel of Fig. 9 depicts the average correlation factor between the expected and predicted potential on the test set as a function of the number of training samples. Focusing only on the blue dots that correspond to a neural network trained with transfer learning and data augmentation with SGM as input, we can observe a small improvement of the prediction

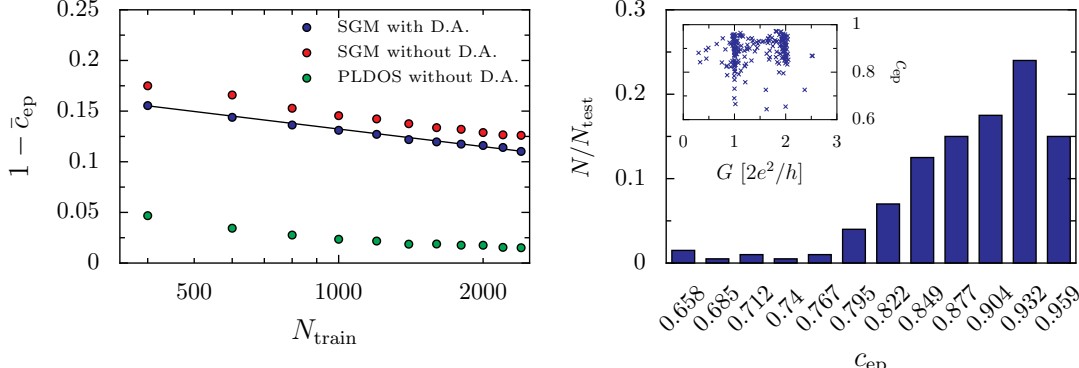

Figure 9: Left panel: evolution of the average correlation factor (over 30 trainings with a different splitting between train and test set) with the training set size. Blue (red) dots represent the cases where the neural network is fed with SGM as input using transfer learning and with (without) data augmentation. The black solid line corresponds to a logarithmic fit with the coefficient -0.305. The green dots correspond to the case where the PLDOS is used as input. In the three cases, the architecture of the neural networks is the same. Right panel: distribution of the correlation prediction on the test set for $N_{\mathrm{train}} = 2450$ and for the case where the SGM is used as input with data augmentation. The inset represents the Pearson correlation coefficient as a function of the conductance (without tip) for all the 200 samples of the test set.

quality with the training set size. Even if we consider a large training set of SGM data with 10.000 samples (requiring an enormous amount of computer time) we cannot expect to reach a much better accuracy than with the current training set composed of about 2500 samples due to the logarithmic evolution of the precision.[11] It is therefore not useful to waste numerical resources to create a huge dataset. However, we can still perform data augmentation that will not lead to a huge improvement of the performance (especially when the dataset is already important), but which is a costless operation. Indeed, the data augmentation (that doubles the dataset) has the same effect as training the neural network with twice the size of the dataset.

The other interesting results concern the difference when the network takes the PLDOS or the SGM as input. Indeed, from Fig. 9, we can state that unveiling the disorder potential from the PLDOS is easier than from the SGM whereas transfer learning has been used only for networks that take the SGM as input. While it has been shown that a neural network can have excellent results (more than 99 % of correlation in average) [18], it is not possible to reach the same accuracy with the SGM. This phenomenon could be related to the loss of locality of the signal with the SGM that is performed with a tip that has a large depletion radius.

# D   Details on the indirect quantitative evaluation of the prediction quality

The indirect evaluation of the neural network performance is crucial for concluding on the reliability of the predicted disorder. The proposed method is based on the relation between the coefficients $c_{i\mathrm{e}}$ and $\bar{c}_i$ defined in Sec. 4.2. Using 20 different disorder configurations and for each of them 4 different SGM maps that correspond to the 4 experimental conductance

---

[11]We consider the logarithmic evolution of the precision in a range that goes to reachable dataset size and not infinity.

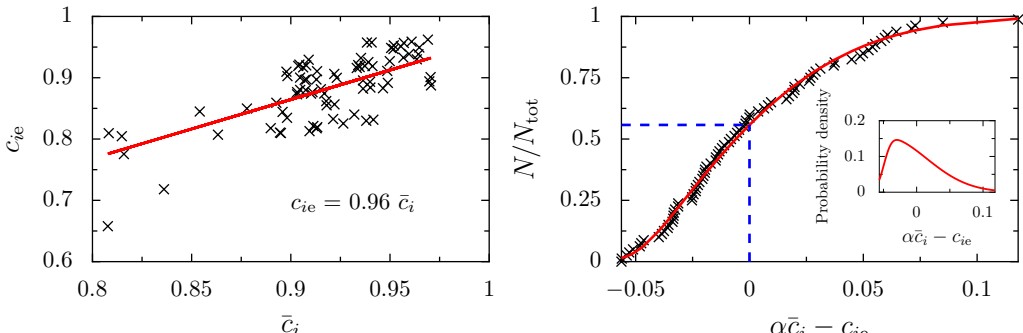

Figure 10: Left panel: the correlation of the predicted potential with the exact one for 20 different simulated samples and four different gate voltages, plotted versus the average correlation with the prediction for the three other gate voltages (see Fig. 5 for an example). The red line represents a linear fit $c_{ie} = \alpha\bar{c}_i$ to the data with $\alpha = 0.96$. Right panel: cumulative density function of the deviation between the real correlation coefficient $c_{ie}$ and the one obtained by averaging $c_{ij}$. The black crosses represent the distribution obtained from the first 20 samples of the test set. The red curve represents the cumulative skew normal function $\gamma$ of Eq. (D.2) with $x = \alpha\bar{c}_i = c_{ie}$ and parameters $\zeta = -0.049$, $\omega = 0.064$, and $\theta = 6.996$ that fits the data. The inset represents the corresponding skew normal function. The dashed blue line represents the point of the fit where $\alpha\bar{c}_i = c_{ie}$.

values, we obtain the 80 $(c_{ie}, \bar{c}_i)$ points[12] presented in the left panel of Fig. 10. From the latter figure, it is possible to estimate the quantity $c_{ie}$ from $\bar{c}_i$ by fitting the data (red solid line). The linear fit indicates that

$$c_{ie} = \alpha\bar{c}_i, \tag{D.1}$$

where $\alpha = 0.96$, meaning that our verification method has a tendency to overestimate the actual Pearson coefficient. This phenomenon can be due to the absence of branches in a given area of the scanning zone, no matter the gate voltage of the QPC. Indeed, in the absence of SGM signal in a given region, one expects that the precision of the disorder prediction decreases in that zone [18]. Therefore, the prediction can be the same when passing SGM data for different gate voltages through the neural network, but it is also different from the real disorder. We note that with the simulated data, we have an average value $\bar{c}_{ie} = 0.89$. The skewed distribution of the predicted correlation around the real one (see Fig. 10) is very well fitted by the skew normal distribution

$$\gamma(x; \zeta, \omega, \theta) = \frac{1}{\omega\sqrt{2\pi}} e^{-\frac{(x-\zeta)^2}{2\omega^2}} \left[ 1 + \text{erf}\left( \theta \frac{x-\zeta}{\omega\sqrt{2}} \right) \right], \tag{D.2}$$

where $\text{erf}(x)$ is the error function. From Fig. 10, we observe that for 84 % of the samples, the deviation from the predicted correlation is below 5 %. The inset of Fig. 10 depicts the associated probability density function.

The presented method establishes an estimation of the correlation coefficient between the predicted potential and the exact one $c_{ie}$ that is obtained from the correlations between different predicted potentials $\bar{c}_i$. The latter quantities are available for the experimental SGM data,

---

[12]Those points are the results of neural network predictions for different splittings between the training and test set, which leads to slightly different results. A correlation between the two quantities $c_{ie}$ and $\bar{c}_i$ is obtained within different neural networks (study performed over 40 different neural networks and the 20 disorder configurations) [25]. Hence, for the rest of the study, we retain the results from the neural network that gives the highest $\bar{c}_i$ coefficient over all the 40 neural networks.

where the exact potential is unknown. The application to experimental data allows us to get the reliability of the predicted potentials that is presented in Sec. 4.

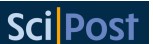

Figure 11: The top line shows experimental SGM data for four different QPC conductance values. Below are examples of images obtained after the application of Gaussian filters with different width on the reconstructed SGM response.

# E   Gaussian filter examples

Examples of simulated SGM maps computed from a predicted potential of the real sample are depicted in Fig. 11, where each line (except the first one) corresponds to a different width of the Gaussian filter. We observe that blurring with a width $\sigma \approx 6\,\text{nm}$ seems to be the most realistic choice.

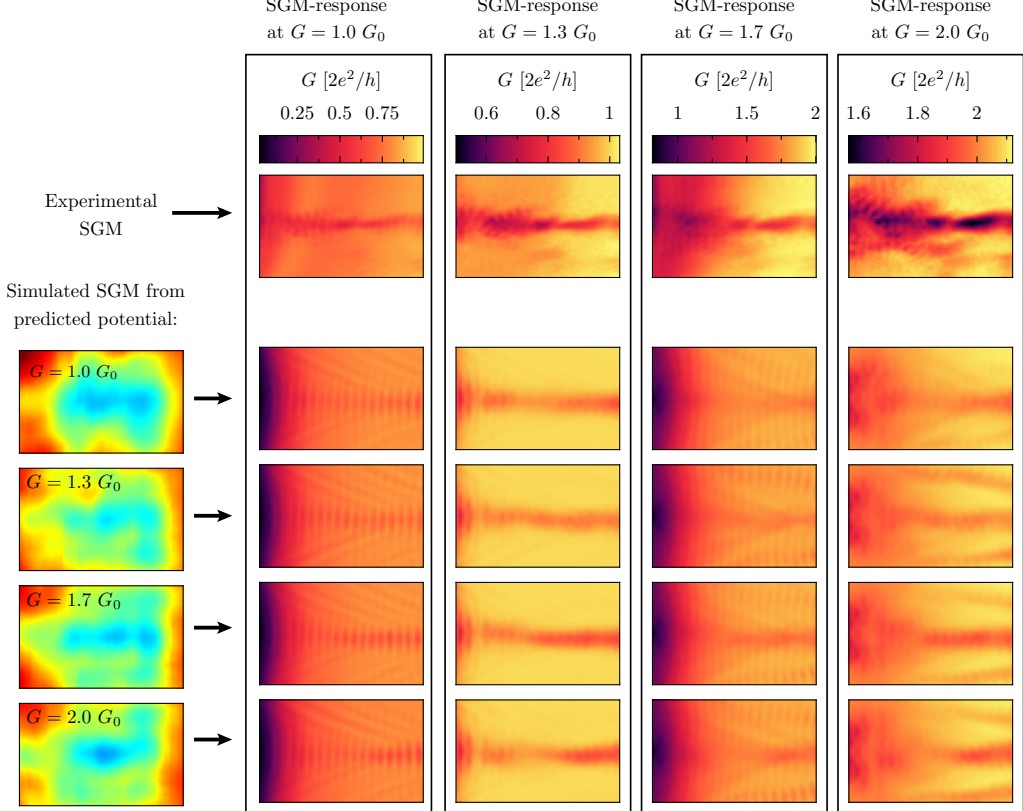

Figure 12: The top line represents the four experimental SGM maps obtained at the indicated QPC conductance values. The four lower lines represent the SGM responses reconstructed from the potentials predicted from the SGM measured at the conductance $G$ indicated in the left part of the figure. A Gaussian filter of $\sigma = 6\,\mathrm{nm}$ is applied on all the simulated SGM maps. Each column corresponds to a different value of unperturbed conductance and has a different color code indicated on the top of the column.

## F Qualitative validation based on reconstructed SGM responses

It is also possible to compute the SGM signal from the predicted potential and to compare it to the original experimental data. To avoid to be skewed by the model used to compute the quantum transport properties, one can compute the SGM in a different experimental condition (e.g. adding a magnetic field or changing the gate voltage) and compare it to the experimental data corresponding to this new experimental condition.

However, no obvious correlation is observed between the similarity of the potentials (the predicted one and the expected one) and the similarity of the SGM images (the SGM computed from the predicted potential and from the expected potential), both evaluated through the Pearson correlation coefficient (2). The hypotheses to explain these results are: (*i*) The lack of information on the disorder between the QPC and the scanning zone (for the SGM reconstruction, we consider no potential fluctuation due to the disorder there) that can lead to a large difference between the original SGM signal and the reconstructed one, even if the predicted potential is correct. And (*ii*), the Pearson coefficient compares two images pixel per pixel, meaning that a small shift of a branch or a phase difference of the interference fringes can lead to a significant decrease in the correlation coefficient. Thus, the value of the coeffi-

cient is mainly dependent on much less relevant features like the cross-talk.[13] This evaluation method can not give any quantitative information but can be considered qualitatively.

From each of the predicted potentials, we computed the SGM for the four different values of $G$.[14] The results are depicted below the associated experimental SGM-response in Fig. 12. Even if the lack of information on the disorder outside of the scanning zone can lead to deviations in the reconstruction, we do not observe a significant impact of the disorder that deviates the electron flow.

We note that despite the difference of the predicted potentials, all the reconstructed SGM maps corresponding to the same conductance value are very similar. Thus, we can suggest that the variation between the predicted potentials has no important impact on the branch pattern (except for the upper right part of the potential from $G = 1.3\ G_0$ where a difference between the reproduced SGM at $G = 2.0\ G_0$ and the real one appears).

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
