# Peer review of "Reconstructing the potential configuration in a high-mobility semiconductor heterostructure with scanning gate microscopy"

_SciPost Physics, doi:SciPost Phys. 15, 242 (2023)_

## Round 1 · Referee Report · Anonymous (Referee 1) · 2023-9-22

Report

Gaëtan et al. present a study wherein they utilize an encoder-decoder neural network (NN) to infer potential configurations from experimental scanning gate microscopy (SGM) data. Their approach integrates PTFS self-consistent calculations and KWANT to generate SGM maps and consequently, the training data for the NNs. An intriguing improvement is the adoption of transfer learning, where they initiate with a PLDOS pre-trained NN. This reportedly overcomes challenges related to data acquisition for training.

In terms of validation, the authors change only the QPC gate voltage, maintaining other parameters unchanged. While this results in changes in the SGM images, the predicted potentials remain largely invariant, providing indirect evidence of their method's physical robustness. An interesting observation they make is the beneficial effect of applying a spatial Gaussian filter to the data, which seemingly enhances reliability.

The findings could potentially pave the way for numerous scanning microscopy experiments to extract spatially resolved local physical quantities using similar techniques. Given its novelty, I believe this work is suitable for publication in SciPost Physics, contingent on addressing the below concern:

Regarding the inverse problem-solving from potentials to SGM measurements through the NN: How well does this method maintain locality? For instance, if Fig.2 (left) were confined to a 100nm x 100nm region, how would both the simulation and NN predictions be affected? A test of this nature would provide valuable insights into how the NN works.

Requested changes

  1. A general discussion about what physically quantity that SGM is measuring can help improve the readability.
  2. In general, I think SciPost does not have a mandatory request for code avalibility. But I think open-source sample code and model can help reader better understand the achievements that the authors made, and increase the reproducibility. Thus, I kindly request the open-source code/data of this study.

  • validity: high
  • significance: high
  • originality: high
  • clarity: high
  • formatting: excellent
  • grammar: excellent

Author:  Dietmar Weinmann  on 2023-11-10  [id 4106]

(in reply to Report 1 on 2023-09-22)
Category:
remark
answer to question

At the end of the report, the referee asks about the locality of the method. The method is expected to work well when the SGM area is large enough to include considerable spatial variations of conductance due to the displacement of the tip. The length scale of the SGM conductance features was investigated in detail in Ref. [11], where it was found experimentally and theoretically that it strongly depends on the tip strength. At strong enough tip as it is used for the production of the SGM data we use in our paper, the SGM structures appear on scales shorter than the Fermi wavelength. A condition for a good quality of the potential prediction is therefore that the size of the scan area is well beyond the Fermi wavelength. Moreover, the SGM response does not only depend on the potential in the vicinity of the tip, but also on more distant potential features, in particular the ones located between the QPC and the tip position. Nevertheless, the fact that the potential is relatively smooth on the scale of the Fermi wavelength places us close to the semiclassical regime in which a more local behavior can be expected.

The requested changes in the report are:

  1. "A general discussion about what physically quantity that SGM is measuring can help improve the readability."

In SGM the conductance of the sample is measured as a function of the position of a charged tip above the surface of the sample. We have extended the description of SGM in the introduction to follow this recommendation.

  1. "In general, I think SciPost does not have a mandatory request for code availability. But I think open-source sample code and model can help reader better understand the achievements that the authors made, and increase the reproducibility. Thus, I kindly request the open-source code/data of this study."

We agree that a well-documented and easy-to-use open-source code is highly desirable. The Kwant package used for the tight-binding transport simulations is available as an open source code since many years. The python package Pescado that we use to solve the electrostatic problem including Thomas-Fermi screening of the conduction electrons is in a very advanced state and will be publicly released as open source code within the next months. Concerning the machine learning parts, the code is not yet general enough nor yet documented enough to allow for a useful release in the present form. We nevertheless aim at an improvement of that situation and hope that a useful form of the code can be released in the future.

---

## Round 1 · Referee Report · Anonymous (Referee 2) · 2023-9-26

Strengths

  1. It presents an approach for estimating the disorder background potential of quantum constrictions based on scanning gate measurements and a machine learning approach.
  2. It presents details of the computations that allows the simulations to be reproduced.
  3. It offers a comprehensive discussion about the techniques and approaches used.

Weaknesses

  1. It only uses one machine learning technique. There may be techniques that offer better results, but this topic is not explored.
  2. The technique of transfer learning is used. Although, it is probably the best that can be done, it offers no guarantees that the estimated potential are close to those one would obtain experimentally.

Report

This is a timely contribution to the field and shows details of an approach for estimating the disorder potential experienced by electrons in a quantum constriction. This estimation is particularly important for designing quantum effect devices, where small variations in the disorder potential can lead to drastic changes in the properties of the device. I recommend the publication of this work and propose a few additions to the paper to make it stronger.

Requested changes

  1. Hafnium oxide has piezoelectric properties. A tip scanning its surface may induce a piezoelectric polarization of the layer, which may affect the background potential. The authors could add a brief discussion if this is really the case, and how much the potential could be affected.
  2. In page 4, the authors say that the "Fermi wavelength ... has been experimentally determined". How was it experimentally determined?
  3. What kind of calculations does KWANT do? In this sense, it would be useful to know where the edge of the first Brillouin zone is compared to the Fermi wavelength. This can lead to limitations in several computational techniques such as tight-binding.
  4. The calculations are performed for a single electron. Therefore, the estimated potentials should correspond to potentials that a single electron would experience. However, it is not made explicit in the discussions if the transport is actually a result of a single electron transport. It would be helpful to add some discussion that clarifies this point.
  5. Also, related to the previous point. It would be helpful to the reader if the authors provided some comparison to potentials experimentally measured elsewhere. Are the estimated potentials similar to the experimentally expected potentials?
  6. Another point related to the previous: After reaching the end of the paper, I had a feeling that I could have learned something from the estimated potentials. What can be extracted from this information? Does it help in designing new devices? How? etc. What does one gain knowing the potential profile of the background?
  7. In page 11 the authors say: "data already yields good results". What does good mean? A qualitative description would be more useful.
  8. It would be very helpful to the reader to see a plot of the quality of training as a function of the number of training samples. This would convey a broader idea about the limitations of the technique. For example, it would indicate if the technique could be improved with a larger training set.

  • validity: high
  • significance: high
  • originality: good
  • clarity: high
  • formatting: good
  • grammar: excellent

Author:  Dietmar Weinmann  on 2023-11-10  [id 4105]

(in reply to Report 2 on 2023-09-26)
Category:
remark
answer to question

Report 2 contains a list of requested changes. We have taken them into account in the revision of our paper. Below are our remarks point by point:

  1. "Hafnium oxide has piezoelectric properties. A tip scanning its surface may induce a piezoelectric polarization of the layer, which may affect the background potential. The authors could add a brief discussion if this is really the case, and how much the potential could be affected."

The tip never touches the surface of the sample, therefore a piezoelectric polarization is not expected to occur. We have mentioned this point in a new footnote in Sec. 2.1 of the revised manuscript.

  1. "In page 4, the authors say that the "Fermi wavelength ... has been experimentally determined". How was it experimentally determined?"

The Fermi wavelength is inferred from the periodicity of the spatial conductance oscillations observed in SGM scans. We included this information in Sec. 2.1 of the revised paper.

  1. "What kind of calculations does KWANT do? In this sense, it would be useful to know where the edge of the first Brillouin zone is compared to the Fermi wavelength. This can lead to limitations in several computational techniques such as tight-binding."

The KWANT package allows to calculate electronic properties within a tight-binding model approximation, including the linear conductance of the sample. For a good description it is indeed important that the lattice constant remains reasonably small as compared to the Fermi wavelength. In our case the ratio is about 1:8. We have mentioned this value explicitly at the beginning of Sec. 3.

  1. "The calculations are performed for a single electron. Therefore, the estimated potentials should correspond to potentials that a single electron would experience. However, it is not made explicit in the discussions if the transport is actually a result of a single electron transport. It would be helpful to add some discussion that clarifies this point."

The calculations take into account electron-electron interactions on the level of the Thomas-Fermi screening, leading to an effective single-electron picture. Electron-electron correlations beyond this level of approximation become important in particular situations, like low dimensionality and low electron density. An example is the 0.7 anomaly on the transport through QPCs. When the regime of the 0.7 anomaly is avoided, and provided the disorder potential remains well below the Fermi energy, as in the case of our paper, the SGM in the vicinity of a QPC is quite well described by effective single-particle simulations. For similar samples containing QPCs in two-dimensional heterostructures, this expectation is confirmed in Ref. [3]. We explain this issue in the revised manuscript at the end of Sec. 2.

  1. "Also, related to the previous point. It would be helpful to the reader if the authors provided some comparison to potentials experimentally measured elsewhere. Are the estimated potentials similar to the experimentally expected potentials?"

Unfortunately, the disorder potential in semiconductor heterostructures containing high-mobility two-dimensional electron gases is not directly accessible, and we are not aware of any measured potential in this kind of sample. The fact that the disorder potential is not known experimentally is the main motivation for our work, as mentioned in the introduction and the conclusion of our paper.

  1. "Another point related to the previous: After reaching the end of the paper, I had a feeling that I could have learned something from the estimated potentials. What can be extracted from this information? Does it help in designing new devices? How? etc. What does one gain knowing the potential profile of the background?"

At present, many mesoscopic transport experiments have an enormous drawback, namely that they are not reproducible and not quantitatively predictable because of the sample-to-sample fluctuations of the disorder configuration. A method that allows to determine the disorder potential allows to obtain the missing information to perform quantitative simulations. This is important to test our understanding of transport in such devices beyond the qualitative level. Moreover, the development of device fabrication methods could become more controlled when the potential landscape becomes accessible and not only the electronic properties. We added two sentences at the end of the outlook of Sec. 5 to explain this point.

  1. "In page 11 the authors say: "data already yields good results". What does good mean? A qualitative description would be more useful."

This sentence appears in the Conclusions of Sec. 5 and refers to the disorder prediction without the blurring data augmentation. As shown in detail in Sec. 4, the precision of that prediction is 67 %, while the data augmentation allows reaching a precision of 87 %. The precise meaning of the sentence "already good results" is therefore "a precision of 67 %". We believe that the qualitative formulation in the conclusions is more readable than to repeat the precise values that are clearly stated in the previous section.

  1. "It would be very helpful to the reader to see a plot of the quality of training as a function of the number of training samples. This would convey a broader idea about the limitations of the technique. For example, it would indicate if the technique could be improved with a larger training set."

Such a plot is presented in Appendix C, in the left panel of Fig. 9. To draw more attention to this figure, in the revised version we mention that figure also in the main text, in the first paragraph of Sec. 4.

---

## Round 1 · Referee Report · Anonymous (Referee 3) · 2023-10-20

Strengths

1-High innovative potential with a substantial interdisciplinary component

2-Clear concept

3-May become important for device fabrication

Weaknesses

Since this is a quite new approach, I would not rate still existent open questions as really weak points. The paper more likely points out a way towards establishing a new characterization method that, of course, has still room for improvements (see report). This kind of weakness may also be rated as a strength by possibly opening up a new field for the community to deal with.

Report

The Authors aim at using SGM maps that consist of the measured conductance response of an electronic system depending on the lateral position of an SGM tip. Simply speaking, the SGM tip probes the local properties of the electron system that finally should provide information of the underlying lateral distribution of the random disorder potential.

Because of the ongoing minimization of electronic devises the development of such a method for determination of the individual fingerprint of potential fluctuations is of major importance. In the case that the size of the devices get of the order of the size of individual fluctuations, a pure statistical representation of disorder may no longer be correct.

Although there can be expected a causal relation between a particular lateral disorder potential distribution and a particular lateral conductance response pattern caused by the SGM tip position, it is hopeless to look for a physical model that directly maps lateral conductance patterns onto the lateral disorder potential distribution. Therefore the authors propose and apply machine learning to overcome that challenge.

However that challenge becomes even bigger because of the fact that there are no experimental training data available for the learning algorithm, which would require to have that problem already solved for at least some particular real cases.

For the training of the learning algorithm the Authors therefore use simulated transport data that result from particularly designed artificial (known) disorder potentials. For getting the training data they have to manage two major tasks: While task one consist of modelling the position dependent impact of the SGM tip on the electronic system by solving the Schrödinger – Poisson problem self consistently, the second task is generating the transport data. The latter is based on the widely used software package KWANT, while the first task is laid out in detail in in previous work of some of the authors.

On the background of the above very briefly summarized concept I can say that the paper is innovative, clearly written and the description of the different tasks is scientifically sound.

Already at this point I would highly recommend publication, although there still can be found reasons for criticism, as it is mostly the case if introducing new methods with high innovative potential. However, we should not forget that one of the major missions of a publisher is to contribute to a constructive discourse within the scientific community and at least to some extend the readers should get their own chance for criticism. Therefore still open questions should not prevent publication in the first place, as long as all ingredients are laid out correctly.

At this point I would like to address one of those possibly still open questions on my own. In task one the authors address screening on the Thomas – Fermi and/or Hartree level based on previous work. In a bench mark test in that previous work they address the IQHE problem and end up with results for the potential and carrier distribution at the edge that agree very well with a famous and highly cited publication of D. B. Chklovskii, B. I. Shklovskii and L. I. Glazman (Ref.36). That gives the authors confidence that they address the electronic system correctly. However, about 10 years ago Pascher et al (DOI10.1103/PhysRevX.4.011014) addressed the edge channel transmission of QPCs in the QHE regime via SGM and found out that they cannot successfully model their transmission data based on a modelling relying also on Thomas-Fermi screening. They argue that in such a model screening might be considerably over estimated. Some most recent citations of that Pascher paper seem to challenge the picture of Chklovskii, Shklovskii and Glazman, which finally might have an impact also on this paper.

Since machine learning has nothing to do with the underlying physics, training with data from a physically wrong model would always allow to regain the correct underlying artificial test potential. The application to a real experimental SGM image then would also result in a particular disorder potential. However, that resulting image might still not have much to do with the real potential because of looking at it with the “wrong eyes”.

However, there is a point that may be in favour of the author’s model, namely the fact that the QHE regime uses of course high magnetic fields that might modify interactions considerably. The authors in this paper use their model without magnetic field, which might justify the Thomas-Fermi approach to be a sufficient approximation. Maybe the Authors want to add a comment on this?

Anyway, I think the Authors did the best that could be done at this stage. They even tried some sort of modifying the real experimental conditions (such as e.g. different openings of the QPCs) to eliminate side effects, since these should not affect the predicted potentials. As I already said, despite the fact that there still can be found open questions, the paper should be made public in order to fuel the discussion within the scientific community and stimulate further developments.

On this background I even tend to recommend publication as it is, unless the Authors take the option of making a comment as mentioned above, which, of course, is not obligatory from my side!

Requested changes

Like stated above, there is room for improvements of the method and to some degree that should also come from the scientific community. Therefore I find it much more important to publish as soon as possible instead of digging out further things that probably still could have been done in the paper.

Therefore no changes required from my side.

  • validity: high
  • significance: top
  • originality: high
  • clarity: high
  • formatting: excellent
  • grammar: excellent

Author:  Dietmar Weinmann  on 2023-11-10  [id 4104]

(in reply to Report 3 on 2023-10-20)
Category:
remark
answer to question

Based on the results presented in E. Chatzikyriakou et al., Phys. Rev. Research (2022) (Ref. [3] of our paper), we have a high degree of confidence in the accuracy of the Thomas-Fermi screening approximation when no magnetic field is applied to the system. However, we agree that our model has limitations when a magnetic field is present, as the Thomas-Fermi approximation is no longer sufficient to accurately describe the system. In our publication, we did not use any magnetic field but we propose to use it for further studies. Indeed, the magnetic field would allow us to perform experiments in very different experimental conditions which can be important for the validation of the disorder prediction. The QHE regime is however not expected to be useful for the disorder determination from SGM data since the transmission of edge channels is neither affected by weak disorder nor by the presence of the tip. The application of a weak magnetic field has to be seen as an important tool but which has to be carefully used, specifically in the simulations.

The remark concerning the possible bias of the neural network to simply inverse the quantum transport simulation leading to a result that can be different to the one actually present in the sample is an important point. As mentioned by the referee, our work is a first step towards the development of a method to determine the disorder present in a heterostructure where we propose methods to tackle the different difficulties that appear. To increase the confidence in the disorder that is predicted by the neural network, we propose to compute the SGM from a predicted disorder within different experimental conditions and to compare it to the corresponding experimentally measured data. The success of such a consistency check makes the presence of a systematic bias due to limitations of the simulations of training data more unlikely. A way to solve such a bias issue could be an iterative process between the improvement of the reliability of the neural network predictions and the improvement of the quantum transport simulations. We added a new paragraph towards the end of the conclusion and outlook of Sec. 5 to address this issue.

---

## Round 2 · Referee Report · Anonymous (Referee 5) · 2023-11-13

Report

I appreciate the authors' comprehensive consideration of all suggestions from the referees and their adequate response to them. This work is significant, and I recommend its publication in SciPost Physics.

I would like to hear the authors' thoughts on the issue of 'locality' that I raised in the last round of review. Although I acknowledge that it might be beyond the scope of this work and a definitive answer is not obligatory, it remains an interesting point:
Specifically, if the SGM of an area A produces a random potential related to A, and the data is then cropped into a sub-area B⊆A (potentially close to a feature-rich side), will the machine learning algorithm still produce an accurate random potential for the cropped area B⊆A? If not, how can this violation of locality be understood?

  • validity: top
  • significance: top
  • originality: high
  • clarity: top
  • formatting: perfect
  • grammar: perfect

Author:  Dietmar Weinmann  on 2023-11-29  [id 4158]

(in reply to Report 2 on 2023-11-13)
Category:
answer to question

The referee asks about the locality of the relation between the disorder potential and the SGM-response. Following the remark, we have investigated the dependence of the disorder prediction quality on the size of the SGM scan. The results of that study are presented in the attached file.

Attachment:

precision_dependence_on_scan_area.pdf

---

## Round 2 · Referee Report · Anonymous (Referee 4) · 2023-11-13

Report

The authors have significantly improved the manuscript and incorporated all the requests made by the referees. It is my understanding that the manuscript meets all the acceptance criteria for publication. This is a timely contribution to both mesoscopic physics as well as to the use of machine learning techniques for solving inverse problems.

---

## Round 2 · Referee Report · Anonymous (Referee 6) · 2023-11-21

Report

I appreciate the effort of the authors to address the points raised in my previous report and the reports of the other referees. This adds to the high quality of the paper.
I recomment publication in the present form.

---

## Round 2 · Author Response

Following the editorial recommendation, we have replied to the three reports on our work, and we have revised the manuscript taking into account the comments made by the referees. We herewith resubmit our paper "Reconstructing the potential configuration in a high-mobility semiconductor heterostructure with scanning gate microscopy" for publication in SciPost Physics.

---

## Round 2 · List of Changes

- We have extended the description of SGM in the introduction.

- We have added a new footnote (2) in Sec. 2.1, stating that the tip never touches the surface of the sample.

- We have added information about the experimental determination of the Fermi wavelength in Sec. 2.1.

- We have mentioned the value of the ratio between the Fermi wavelength and the tight-binding lattice constant in Sec. 3.

- A remark about the limited importance of electronic correlations has been added at the end of Sec. 2.

- In the first paragraph of Sec. 4, we mention Figure 9 that shows the training-set size dependence of the accuracy.

- The risk of possible general bias of the method mentioned by referee 3 is discussed in the new second-to-last paragraph in Sec. 5.

- At the end of Sec. 5, a new paragraph mentions some possible uses of the proposed method.

- We have corrected a few typos. We have also updated the list of references and funding information.

---

## Editorial Decision

published